

# Sinding-Larsen-Johansson disease. Clinical features, imaging findings, conservative treatments and research perspectives: a scoping review

Bartosz Wilczyński[1], Marcin Taraszkiewicz[2], Karol de Tillier[2], Maciej Biały[3] and Katarzyna Zorena[1]

[1] Department of Immunobiology and Environmental Microbiology, Medical University of Gdansk, Gdansk, Poland
[2] Medical University of Gdansk, Gdansk, Poland
[3] Institute of Physiotherapy and Health Sciences, The Jerzy Kukuczka Academy of Physical Education, Katowice, Poland

## ABSTRACT

**Objective:** This review aims to consolidate existing research on the pathogenesis, clinical diagnosis, imaging outcomes, and conservative treatments of Sinding-Larsen-Johansson disease (SLJD), identifying literature gaps.

**Design:** Scoping Review.

**Methods:** A comprehensive literature search was conducted across databases including PubMed, Scopus, Medline OVID, Embase, Web of Science, and Grey literature following the Preferred Reporting Items for Systematic Reviews and Meta-Analyses (PRISMA) extension for scoping reviews. The quality of included studies was assessed using the Joanna Briggs Institute (JBI) checklist.

**Results:** The body of evidence on SLJD, primarily derived from case studies, reveals limited and often conflicting data. Key findings include: (1) SLJD commonly presents as localized knee pain in physically active adolescents, particularly males, (2) ultrasound and MRI are the most effective diagnostic tools, (3) conservative treatment, which mainly focuses on activity limitation, yields positive outcomes within two to eight months.

**Conclusions:** Our review shows that SLJD mainly affects physically active adolescents aged 9–17 years. The authors recommend conservative treatment, rest and/or cryotherapy, passive mobilization, muscle restraint, isometric exercise, and NSAIDs. Further cohort studies are necessary to refine the management and application of the SLJD treatment database.

Corresponding author
Bartosz Wilczyński,
b.wilczynski.fizjoterapia@gmail.com

# INTRODUCTION

## Injury issues in children and adolescents

Over the decades, physical activity among adolescents has shifted from casual participation in sports or physical work to organized, regular, and controlled sports

activities (*Hawkins & Metheny, 2001*). Moreover, sports participation seems to be increasing in recent years (*Merkel, 2013*; *Patel, Yamasaki & Brown, 2017*; *LeBrun et al., 2018*; *Danes Daetz, Rojas Toro & Tapia Mendoza, 2020*; *Prieto-González et al., 2021*). According to previous data (*Merkel, 2013*; *Stracciolini et al., 2014*), approximately 45 million children and adolescents in the United States participate in youth sports (*Merkel, 2013*; *Hall et al., 2015*). There has been a consistent increase in sports specialization in last years (*Hall et al., 2015*). This trend is also evident among young women, with a growing number of professional female sports clubs being established (*Stracciolini et al., 2014*). Musculoskeletal injuries, including acute and overuse injuries, are the primary reasons for young athletes visiting family physicians (*Patel, Yamasaki & Brown, 2017*). Young male athletes who train full-time, experience high rates of overuse bone injuries and muscle strains (*Martínez-Silván et al., 2021*). In the United States, more than 2.6 million medical visits for sports-related injuries (ages 5–24) are reported, with 70–80% of cases being attributed to 'sports burnout' before the age of 15 (*Merkel, 2013*). Soccer, basketball, American football, baseball, softball, and track and field have the highest injury rates among young athletes in United States (*Patel, Yamasaki & Brown, 2017*). It is important to consider the specificity of each sport, as well as its popularity and the amount of data published in studies (*Patel, Yamasaki & Brown, 2017*).

## Sinding-Larsen-Johansson disease (SLJD)

Sinding-Larsen-Johansson disease (SLJD) is widely recognized as a traction osteochondrosis affecting the distal patellar region (*Orava & Virtanen, 1982*; *Malherbe, 2019*). Although the amount of quantitative studies and scientific evidence of SLJD is limited, most authors consider the cause of the traction mechanism is similar to Osgood-Shlatter disease (OSD) (*Suzue et al., 2014*; *Patel & Villalobos, 2017*; *Malherbe, 2019*; *Alito et al., 2023*). Both OSD and SLJD involve increased traction of the patellar tendon by quadriceps tightness, and each condition can occur bilaterally (*Ziskin et al., 2023*). SLJD can result from repetitive microtrauma to the inferior patellar pole caused by patellar tendon traction, leading to irregularity and fragmentation of the inferior patellar pole during bone maturation in young patients (*Negrão et al., 2023*). While predominantly affecting adolescents, SLJD is characterized by pain localized at the inferior pole of the patella, often complicating diagnosis due to overlapping symptoms with other knee pathologies. SLJD is generally considered a "self-healing" condition, with most symptoms subsiding after a period of skeletal growth. However, as with OSD, some of the symptoms (pain, tenderness of the patellar tendon, limitation of activity) may remain (*Bruzda, Wilczyński & Zorena, 2023*). The condition can also progress to more severe symptoms, such as patellar sleeve fractures or inferior pole fractures (*Schmidt-Hebbel et al., 2020*; *Devana et al., 2022*). Despite its clinical relevance, research on SLJD is scarce, with most existing studies being case reports.

## Epidemiology of SLJD

SLJD data have mostly been reported in case studies due to the rarity of the condition. However, the last decade has seen an increase in articles describing this entity with greater

incidence. It is possible that we now have more knowledge regarding the classification of anterior knee pain, specifically the inferior pole of the patella, as SLJD (*Gerbino, 2006*; *Davis et al., 2023*). The second potential cause is that a higher percentage of SLJD is associated with the increased popularity of sports among children, sports specialization, and higher loads during sports performance (*Davis et al., 2023*).

In 2014, *Suzue et al. (2014)* investigated the prevalence of osteochondrosis among 547 young soccer players reporting pain. Of these, 106 underwent radiographic and ultrasound examination. The study found that 80 participants (75.5%) showed osteochondrosis, including 13 cases of OSD and 10 cases of SLJD. Therefore, based on the study's data, the prevalence of SLJD was 12.5% among all soccer players who underwent imaging examination due to reported knee pain. *Materne et al. (2022)* conducted a prospective study on 551 youth soccer players over four years, reporting 307 physeal injuries (14% of all injuries). Of these injuries, 85% (258) were apophyseal. The incidence of SLJD was 16, accounting for 6% of all injuries (*Materne et al., 2022*).

Traction osteochondrosis also occurs in the cerebral palsy patients. The incidence of SLJD in this group ranges from 5% to 28% (*Kaye & Freiberger, 1971*; *Rosenthal & Levine, 1977*; *Tyler & McCarthy, 2002*).

## Potential pathogenesis of SLJD

The prevailing theory on the pathogenesis of SLJD, as well as other traction osteochondroses (OSD, Sever's disease), is that the condition is caused by repetitive, prolonged loading, stress, or micro-trauma on the apophysis (*Orava & Virtanen, 1982*; *Launay, 2015*; *Malherbe, 2019*; *Nieto-Gil et al., 2023*). In SLJD, this occurs in the distal area of the patella or the proximal insertion of the patellar tendon (*Rosenthal & Levine, 1977*; *Iwamoto et al., 2009*; *Alito et al., 2023*). The injury and its symptoms are a result of the crossing of the biological barrier and biomechanical properties of the specific bony region. Studies of patients with spastic cerebral palsy suffering from SLJD have supported the theory of repetitive trauma. *Rosenthal & Levine*'s *(1977)* study, which examined 85 cases of cerebral palsy, identified seven cases of SLJD with concurrent OSD. One explanation for this phenomenon, was myofascial spasticity and mechanisms that stress weakened zones in the non-growing cartilage (*Kaye & Freiberger, 1971*; *Rosenthal & Levine, 1977*; *Tyler & McCarthy, 2002*).

*Hall et al., 2015* reported, that sports specialization (focusing on one sport) is a significant factor in the development of anterior knee pain (study among a large cohort, $n = 546$, of young women). The researchers compared women practicing one sport with athletes who play multiple sports simultaneously, sports specialization was found to be associated with a fourfold increased risk of SLJD and OSD (95% CI [1.5–10.1], $P = 0.005$) (*Hall et al., 2015*). A study on factor analysis associated with SLJD found a statistically significant increase in posterior tibial slope in the SLJD group compared to the control group. The authors hypothesized that engaging in running sports at a young age could lead to an increase in posterior tibial slope, resulting in quadriceps overload and the development of SLJD (*López-Alameda et al., 2012*).

The data describing the course of clinical examination, conservative treatment and time to return to activity are disparate and heterogeneous. From a clinician's perspective, managing patients with SLJD according to a scientific model appears complicated. The authors of this review posed the questions: how to diagnose and how to manage conservatively young patients with SLJD from a clinical perspective. Therefore, the purpose of this scoping review was to synthesize the available studies and attempt to clarify conclusions about diagnosis and treatment of the SLJD. To the best of our current knowledge, this is the first review strictly focused of SLJD research. For disease entities belonging to rare diseases, scoping review seems reasonable to help both clinicians and researchers understand the above issues.

## METHODS

### Protocol

The protocol was composed using the PRISMA-ScR (Preferred Reporting Items for Systematic Reviews and Meta-analysis Protocols-extension for scoping reviews) (Fig. 1). Due to discrepancies in terminology, the search strategy included not only the eponym SLJ but also terms such as osteochondrosis and apophysis. This approach allowed for the identification of a greater number of reports.

The protocol aimed to find studies that clinically and/or functionally evaluated young patients with confirmed SLJD without acute trauma, with or without conservative treatment. Studies describing an acute knee injury followed by a diagnosis of SLJD were not included, due to potential diagnostic confusion and an incomplete picture of the chronic condition that is SLJD.

The research questions posed were: (1) What are the consistent symptoms and signs in patients with SLJD? (2) Is there consistency in the imaging diagnosis of SLJD? (3) What are the options for conservative treatment of SLJD? (4) What is the prognosis and the time frame for returning to sports or physical activities? The final protocol is available in the Open Science Framework (https://osf.io/z7cnw/).

### Eligibility criteria

Inclusion criteria covered original studies (including doctoral dissertations and conference abstracts) that evaluated pediatric patients (under 18 years of age) with overuse SLJD, without concurrent injuries. These studies included clinical and/or functional assessments as well as imaging evaluations (MRI, US, or X-rays), with or without conservative treatment, and were written in English or French. The articles included were those with peer-reviewed journals with no restriction on year of publication, from the year (1921) of the first scientific reports (first article described SLJD) to 2023.

Exclusion criteria were as follows: (1) studies involving animals, adult participants, patients with acute knee injuries (including acute trauma and SLJD), or patients with coexisting conditions (such as Osgood-Schlatter disease, Sever's disease, patellar sleeve fractures, or cerebral palsy disorders), and those who underwent surgical treatment for SLJD; (2) review articles; (3) studies from suspected predatory scientific journals (based on

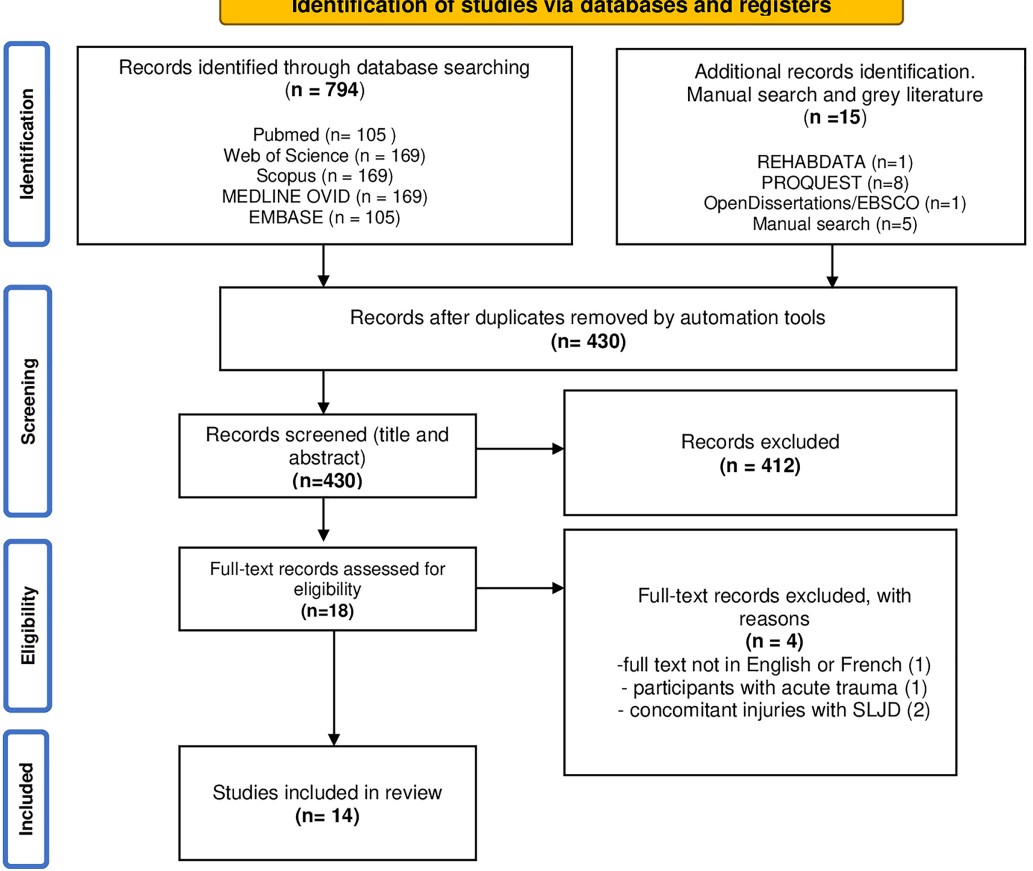

**Figure 1 PRISMA flow diagram.** The selection process of suitable articles included in this scoping review. The process involves initial identification through database searches, screening based on titles and abstracts, full-text assessments, and final inclusion based on predefined criteria.

open academic journal predatory checking system–AJPC) (*Chen et al., 2023*); and (4) studies without available full-text articles.

## Information sources and literature search

A comprehensive database search was performed in December 2023 and subsequently refined and improved on June 29, 2024. To identify relevant literature, the databases PubMed, Web of Science, Scopus, Medline OVID, and Embase were searched. Additionally, gray literature was searched in ProQuest, REHABDATA, and OpenDissertations/EBSCO. A manual search of the reference lists in the included articles was also conducted. Search model were used with terms from Medical Subject Headings (MeSH), free-text terms (TIAB) and keywords (complete list of terms contained in the Supplemental Materials): "sinding*" (TIAB), "Sinding Larsen Johansson" (MeSH), "SLJ syndrome" (TIAB), "patellar injur*" (TIAB), "inferior pole of patella" (MeSH), "traction osteochondr*" (MeSH), "patellar apophys*" (TIAB), "traction apophys*" (TIAB). All terms were searched in the context of clinical features, symptoms, imaging studies, and

pediatric populations. The complete search strategy was contained in the Supplemental Materials (https://osf.io/z7cnw/).

## Reports selection

All authors confirmed the same search database results through research team discussion. All studies identified through the search strategy were screened based on the eligibility criteria. Three independent reviewers assessed each title and abstract to determine whether it met the inclusion requirements. To prevent the problem of duplication from different sources, doubled articles have been removed in the bibliography manager (Mendeley, version 1.19.8) and web application for systematic reviews (Software as a Service, Rayyan) (*Ouzzani et al., 2016*).

Subsequently, reviewers conducted a full analysis of the included publications by evaluating full texts. The entire process was performed using Rayyan software (*Ouzzani et al., 2016*). Data extraction and analysis was discussed in joint meetings. The final decision was reached by consensus; however, if needed, a fourth reviewer assessed the text and made the decision.

## Charting the data and data items

Two independent reviewers extracted and charted all relevant information from the included studies using a data extraction grid to ensure methodological standardization (see Table 1). In case of discrepancies, a third reviewer independently re-verified the table for information content and made the final decisions. The data that were extracted concerned the article characteristics: (1) Author and year of publication of the paper and (2) type of study. Next, information was collected on (3) participants (gender and age), (4) clinical characteristics (measurements and interviews taken by clinicians), and (5) data from available imaging studies including magnetic resonance imaging (MRI), ultrasonography (US), and X-rays. Finally, data on (6) conservative intervention and (7) results (including the time to return to sport) were extracted.

## Critical appraisal of individual sources of evidence

Two reviewers indepedenly evaluated the quality of the included studies (Table S1). Any disagreements between the reviewers during the various stages of the review process were resolved through discussion. The quality of the included case reports was assessed using the Joanna Briggs Institute Critical Appraisal Checklist for Case Reports, which comprises eight yes/no/unclear questions (*Tufanaru et al., 2024*). To summarize the overall quality of the case reports, they were categorized into three groups: (1) Low risk of bias (studies that met at least 75% of the quality criteria), (2) Moderate risk of bias (studies that met between 50% and 74% of the quality criteria), and (3) High risk of bias (studies that met less than 49% of the quality criteria).

## Synthesis

Included and analyzed reports are summarized and described in Table 1. The extracted data were analyzed, categorized, and interpreted to provide a comprehensive overview of the current knowledge related to the research questions and to highlight areas for future

**Table 1 Characteristics of included studies.**

| Authors and year | Study | Participants | Clinical features | US | Diagnostic findings X-Ray | MRI | Intervention | Results |
|---|---|---|---|---|---|---|---|---|
| Sinding-Larsen (1921) | Case reports (two) | 10 and 11-year-old (F) | Clinically unilateral, pain after and during jumping, dancing, painful during "percussion" (on the patella) in one case: non-tender swelling on the patella tendon, full knee range of motion. | | The anterior outline of the right patella appears hazy, with the cortex seemingly destroyed to the apex. In the thickened soft tissue in front of the patella, two or three narrow, oblong calcium or bone shadows are visible near its anterior edge. | | Limiting painful physical activity (in both cases). One case - Application of a plaster of Paris bandage with fenestra on anterior side of knee for 6 weeks. | After 6 weeks: symptoms started to regress (no pain, no swelling) After 6 month, return to sport: Dance in the ballet and go ski-ing. |
| Franceschi et al. (2007) | Case report | 9-year-old competitive skater (M) | Billateral knee pain, Painful activities: ascending the stairs, long-distance walking, running and skating | | X-ray axial image showing areas of fragmentation over the entire surface of the patella. | Fragmentation of both ossification centers of the patella, the physis remains open, there is no damage to the articular cartilage and no microfractures. | Limiting painful physical activity, Off-loading the knee with crutches for 2 months, isometric exercise, and NSAID (no details). After 2 months, from partial to full weigbt bearing (3 –week period). Next 2 months: muscle strengthening and swimming | After 6 months symtoms improve and patient return to sports activity. |
| Valentino, Quiligotti & Ruggirello (2012) | Case report | 13-year-old soccer player (M) | Anterior, spontaneous knee pain (during or after prolonged knee flexion and loading), swelling at the inferior pole of the patella. | Ultrasound revealed cartilage edema, fragmentation of the infrapatellar pole, thickening of the proximal insertion of the patella, and the presence of a serous bursa with fluid between the patella and the infrapatellar pole | | | Limiting painful physical activity | After 5 months: The symptoms rapidly regressed and (the patient had full recovered). US revealed normal findings. |

(Continued)

| Authors and year | Study | Participants | Clinical features | US | Diagnostic findings X-Ray | MRI | Intervention | Results |
|---|---|---|---|---|---|---|---|---|
| Kuehnast, Mahomed & Mistry (2012) | Case report | 12-year-old (M) | Knees pain for 2 months. Patient was diagnosed with acute lymphocytic leukemia on referral from pediatric oncology, which had been in remission for five years. Bilateral mild knee effusions with tenderness over the infrapatellar area. Swelling in both knees. Painful kneeling. | | X-ray: absence of significant peripatellar edema of soft tissue or bone fragments adjacent to the infrapatellar pole. | MRI of both knees, sagittal plane, T2-weighted, with fat suppression. The sequence revealed hyperintense signal in the inferior pole of the patella, proximal and posterior patellar tendon, and surrounding soft tissues | Limiting painful physical activity The patient was prescribed 250 mg of naproxen twice daily. | After 2 months, the patient's symptoms had disappeared. |
| Goldmann (2012) | Case report | 11-year old – horseriding (F) | Pain in the leftknee (after horseback exersies). Full normal knee range of motion. | | | MRI with PD-weighting in the sagittal plane showed an irregular shape and appearance of the bone of the lower pole of the patella with irregular lines giving the impression of "fracture lines". In addition, in all cases of fat deposition, there was edema of the infrapatellar pole and hence the insertion of the patellar tendon. | Limiting painful physical activity stop horsebackexercises and training for half a year. | No info |
| Tourdias & Erésué (2015) | Case report | 11-year old soccer player (M) | The knee pain had been progressing for several months. The apex of the patella was very painful on palpation as well as on isometric testing and quadriceps stretching. | A standard radiograph showed a secondary ossification nucleus at the apex of the patella, with soft tissue swelling | | MRI revealed bone fragmentation and swelling of the lower pole of the patella with thickening of the proximal part of the patellar tendon showing hypersignal extending into Hoffa's fat. | Conservative treatment was based on sports rest, sometimes combined with the use of a splint, and specific rehabilitation, with priority given to stretching the knee extensors | The outcome was positive after functional treatment, with return to sports after 6 months. |
| Alito et al. (2023) | Case report | 10-year-old amateur soccer player (M) | Three month bilateral knee pain (left knee worse). Difficulty walking, climbing and descending stairs and kneeling. Full knee ROM. Tenderness over the patellainferior pole and the patellar tendon. | US showed bilateral infrapatellar pole fragmentation with tendon thickening. Also, cartilage swelling and a serous distnsion of the synovial bursa (infrapatelalr). | Standard X-rays showed bilateral tendons calcification (inferior pole). | | Limiting painful physical activity (8-12 weeks), NSAIDs (ibuprofen) as needed, stretching and isometric exercises for lower limb muscle (for 2 months), and additionally cryotheraphy. | After 2 months, reduction in pain, no difficulty in walking and stairs. US showed withdrawal of inflammation of synovial bursa and realignment of tendon fibers. After 3 months, patient return to activities. |

| Authors and year | Study | Participants | Clinical features | US | Diagnostic findings X-Ray | MRI | Intervention | Results |
|---|---|---|---|---|---|---|---|---|
| *Bonney (1948)* | Case report | 11-year-old soccer player (M) | The patient has experienced aching pain in both knees, predominantly on the left side, for the past eighteen months. The pain was alleviated by rest and exacerbated by exertion. It was particularly severe during the previous winter when he was playing soccer There was no relevant past or family medical history. Physical examination revealed tenderness at the lower pole of the patella, more pronounced on the left side, accompanied by slight swelling in that area. There was no limitation knee range of motion. | | X-rays reveal an accessory ossification center at the lower pole of the patella and fragmentation of the tibial tubercle apophysis, closely resembling the changes seen in OSD. | | Eight weeks of plaster immobilization. | Fusion of the inferior patellar ossicle to the rest of the patella was observed after eight weeks of plaster immobilization. The pain subsided almost completely during the summer but recurred when he resumed playing football this year. |
| *Malherbe (2019)* | Case report | 12-year-old (M) | Persistent infra patellar knee pain during longstanding seated positions and sports. | US of the infrapatellar end showed apophyseal extension with bony irregularities at the tendon insertion site, which looks like apophysitis. Local thickening was noted proximal to the infrapatellar tendon insertion compared to the contralateral side. Doppler showed subtle increased hyperemia in the tendon fibers of the intrapatellar tendon. | Radiographic examination of the symptomatic right knee showed no bony abnormalities at the point of maximum pain and was considered normal. The asymptomatic left knee had cortical irregularities and loose bone fragments. | | No info | No info |
| *De Flaviis et al. (1989)* | Case reports | Two patients, not specified, only as a whole group (10–15 years old, mean, 13), mainly baseball and soccer players | No info | Ultrasound shows the lower pole of the patella appears fragmented and hypoechoic, with swelling of the cartilage, particularly at the insertion of the patellar tendon. | | | No info | After three months, the patients returned to activities. |

(Continued)

| Authors and year | Study | Participants | Clinical features | US | Diagnostic findings X-Ray | MRI | Intervention | Results |
|---|---|---|---|---|---|---|---|---|
| *Iwamoto et al. (2009)* | Case reports | Six boys aged 11-13 years* (M) Sports: soccer/karate/skiing/Kendo | There were tenderness at the inferior pole of the patella in the knees. | | Among the symptomatic knees, some showed regular or irregular calcification at the inferior pole of the patella. In several cases, the calcification had coalesced, while in others, it had incorporated into the patella, resulting in a normal radiographic appearance. In a few instances, a small calcification was observed separated from the patella. | | No info | No info |
| *Davis (2010)* | Case report | 17-year old (M) | Pain symptoms persisted despite skeletal maturity. | | | The sagittal proton density MRI reveals thickening and an increased signal within the proximal patellar tendon just distal to its origin, along with a noticeable intratendinous bone fragment. | No info | No info |
| *Dupuis et al. (2009)* | Case reports | 10-year old (M) | No info | | The X-ray reveals a small area of heterotopic ossification at the inferior pole of the patella. | The T1-weighted MRI shows low-signal-intensity bone marrow edema. | No info | No info |
| | | 14-year old (M) | No info | | The radiograph shows partial fusion of the ossification center. | The T2-weighted MRI displays extensive edema at the ossification site and in the surrounding soft tissues. | No info | No info |
| *Carr et al. (2001)* | Case reports/ Pictorial Essay | 14-year old (M) | Anterior knee "discomfort" | Ultrasound shows thickening of the proximal patellar tendon. The upper patellar surface was smooth, while the distal portion appears irregular. A detached bony fragment is visible within the tendon. | | | No info | No info |
| | | 10-year old (M) | Persistent knee pain and swelling | Ultrasound reveals a swollen proximal patellar tendon and fragmentation of the lower patella, while the upper patellar surface appears normal. | | | No info | No info |

**Note:**
Year, year of publication; US, ultrasound; MRI, magnetic resonance imaging; NSAID, non-steroidal anti-inflammatory drug; M, male; F, female; OSD, Osgood Schlatter disease; *, one patient was excluded due to a concurrent meniscus injury.

research. The articles have been inserted in a table in order of the amount of information they contain. Studies that had no intervention or results were listed at the bottom of the table. Evidence from observational prospective or retrospective studies were described in narrative format in the discussion section.

## RESULTS

### Relevant literature identification

A total of 794 citations from database searching and 15 citations from additional methods (manual serach and grey literature) were identified (Fig. 1), of which automated tools (Ryyan and Mendeley software) found 430 duplicates. Of the 430 titles and abstracts screened, 18 full-text articles were included. Subsequently, four of them were rejected due to exclusion and inclusion criteria (exact reasons described in Fig. 1). Ultimately, 14 studies were included in the review and subjected to in-depth analysis and discussion. In these, we obtained one article written in French. To ensure accuracy, we collaborated with native speakers and used language model-based software to translate the data into English.

### Study characteristics and results

The articles finally included in the review were all case reports with one or more participants described. One study was also a pictorial essay (Carr et al., 2001). Table 1 describes the characteristics of the fourteen included studies, along with data regarding authors, year of publication, brief description of participants, clinical findings, imaging results, interventions performed and outcomes. All articles was published across 1921 and 2023. We identified only eight studies with proposed intervention. All of them had a positive final result of a return to activity/sport. However, recovery time varied strongly between studies. Our scoping review revealed a paucity and limited number of research on SLJD. Therefore, the following sections are described in a combination of a synthesis of the results of 14 case studies and a narrative literature review. Out of the 14 studies, seven were classified as having a low risk of bias, meeting at least 75% of the quality criteria. These studies consistently provided clear descriptions of patient demographics, history, current clinical condition, diagnostic tests, interventions, and outcomes. The remaining seven studies were categorized as having a moderate risk of bias, meeting between 50% and 74% of the quality criteria, often lacking comprehensive details in areas such as patient history, intervention descriptions, and adverse events. No studies were classified as having a high risk of bias.

### Participants

The study encompassed 23 patients diagnosed with SLJD, ranging in age from 9 years (Franceschi et al., 2007) to 17 years (Davis, 2010) (with a mean age of 12 years). Among the participants, three were female. One study (De Flaviis et al., 1989) did not specify the sex or exact ages of the subjects, although it did provide a range and average for the entire group (Table 1). Ethnicity was not documented in the studies. Not all authors reported the sports activities of the participants; however, among those who did, a minimum of five out of 23 patients were engaged in playing soccer.

## DISCUSSION

### Summary of evidence

The aims behind the scoping review were to identify potential clinical features and symptoms, imaging study characteristics, and conservative treatment options for young patients with SLJD. We included 14 case reports that described 23 patients with SLJD. The studies were heterogeneous and inconsistent, differing in the clinical features or imaging patterns description, although they had common denominators, which are described in the sections below.

Moreover, we decided on the search strategy with keywords that focused on broad scope of clinical features, imaging findings, and conservative treatments for SLJD. In our study, we focused on young participants with diagnosed SLJD, no comorbidities (such us cerebral palsy, or OSD) and no consequences of SLJD (such as patella sleeve fractures). To the best of our knowledge, this is the first review of studies on SLJD. Therefore, we consider it useful for orthopedic doctors, physiotherapists and sport specialists in working with SLJD patients.

### Symptoms and clinical diagnosis—clinical features

According to the collected data, mostly affected group are physically active adolescents who participate in sports between the ages of 9–17 (*Bonney, 1948*; *Carr et al., 2001*; *Franceschi et al., 2007*; *Iwamoto et al., 2009*; *Dupuis et al., 2009*; *Davis, 2010*; *Kuehnast, Mahomed & Mistry, 2012*; *Valentino, Quiligotti & Ruggirello, 2012*; *Tourdias & Erésué, 2015*; *Malherbe, 2019*; *Materne et al., 2022*; *Alito et al., 2023*). It seems to affect the male gender more often (*Bonney, 1948*; *Carr et al., 2001*; *Franceschi et al., 2007*; *Iwamoto et al., 2009*; *Dupuis et al., 2009*; *Davis, 2010*; *Kuehnast, Mahomed & Mistry, 2012*; *Valentino, Quiligotti & Ruggirello, 2012*; *Tourdias & Erésué, 2015*; *Malherbe, 2019*; *Materne et al., 2022*; *Alito et al., 2023*). Patients with SLJD report spontaneous, insidious pain, without sudden trauma, with an onset near the upper pole of the patellar tendon. Pain may be progressive to chronic in nature (more than 3 months), in one case (*Bonney, 1948*) it persisted for up to 18 months. Pain was associated with sports activities (*Sinding-Larsen, 1921*; *Franceschi et al., 2007*; *Valentino, Quiligotti & Ruggirello, 2012*; *Malherbe, 2019*), antalgeic gait (even with the limp), climbing and descending stairs (*Franceschi et al., 2007*; *Alito et al., 2023*), kneeling, (*Kuehnast, Mahomed & Mistry, 2012*; *Alito et al., 2023*), and long-distance walking or running (*Franceschi et al., 2007*). In one study (*Malherbe, 2019*), a patient complained of knee pain during prolonged sitting. Pain can occur bilaterally (*Kuehnast, Mahomed & Mistry, 2012*; *Alito et al., 2023*), although it can be with an indication of worse symptoms in one knee. On the other hand, in one of a few case-control studies, of 14 patients, only one had bilateral symptoms (*López-Alameda et al., 2012*). The authors sometimes describe visible (*Valentino, Quiligotti & Ruggirello, 2012*) swelling in this area, although not always (*Sinding-Larsen, 1921*). Palpation examination might reveal tenderness around the inferior pole of the patella (in eight patients) (*Bonney, 1948*; *Iwamoto et al., 2009*; *Alito et al., 2023*) or the apex of the patella (one case) (*Tourdias & Erésué, 2015*). A single case of a 17-year-old boy in the 2010 Davis study showed persistent

symptoms in the patella despite skeletal maturity (*Davis, 2010*). The range of motion of the knee joint in most cases is full, not limited. The clinical features of this entity are nonspecific due to the variety of symptoms described in the cases. There are no specific tests available for diagnosis. However, one author of SLJD described "tenderness on percussion of the whole of the anterior surface of the patella" (*Sinding-Larsen, 1921*). One case study described pain in a patient while tensing the quadriceps of the thigh (*Tourdias & Erésué, 2015*).

Studies worth citing at this point, however, were not included in the review due to exclusion criteria include the study by *Devana et al. (2022)*. The authors studied a large group (*n* = 58) of young patients (10.3 ± 1.4 years) with SLJD, however, 24% of them had acute trauma. Patients presented with knee pain, swelling, and tenderness at the interior pole of the patella, were able to bear weight (88%) and perform a straight leg raise (98%) (*Devana et al., 2022*). Furthermore, patients in the cohort of the *López-Alameda et al. (2012)* study showed short hamstring tendons. Interestingly, SLJD in some patients occurs simultaneously with conditions of similar etiology (Osgood Shlatter or Sever diseases) (*Rosenthal & Levine, 1977*; *Bruzda, Wilczyński & Zorena, 2023*).

### Imaging studies—What can we see?

#### X-ray

X-ray is the most commonly used imaging study for diagnosing SLJD cases, particularly in instances of knee trauma among children and adolescents, to rule out avulsion fractures or patellar fractures. In the study by *Alito et al. (2023)*, both symptomatic knees of a 10-year-old boy were examined with X-rays, revealing calcification of the patellar tendon (lower pole). In the study by *Dupuis et al. (2009)*, X-rays showed a small area of heterotopic ossification at the inferior pole of the patella in a 14-year-old boy with a more severe form of Sinding-Larsen-Johansson syndrome, indicating partial fusion of the ossification center. *Bonney (1948)* noted that X-rays revealed an accessory center of ossification at the lower pole of the patella, along with fragmentation of the tibial tubercle apophysis, resembling changes seen in Osgood-Schlatter disease. *Tourdias & Erésué (2015)* reported that a standard radiograph displayed a secondary ossification nucleus at the apex of the patella with associated soft tissue swelling. Conversely, *Malherbe (2019)* found no abnormalities in standard X-rays of the symptomatic knee. Similarly, *Kuehnast, Mahomed & Mistry (2012)* reported a case where X-rays showed no bony fragments or soft tissue swelling. *Franceschi et al. (2007)* described X-ray axial images showing areas of fragmentation across the entire surface of the patella. *Sinding-Larsen (1921)* observed that the anterior outline of the right patella appeared hazy, with the cortex seemingly destroyed down to the apex. Additionally, two or three narrow, oblong calcium salt or bone shadows were visible close to the anterior outline within the thickened soft tissues in front of the patella (*Sinding-Larsen, 1921*). *Iwamoto et al. (2009)* observed differences in radiographic evaluations of eight knees (seven young male athletes, 11–13 years old), noting that abnormal findings around the lower pole of the patella may have different etiologies, making it difficult to differentiate between tendinitis, stress fractures, and SLJD from an X-ray perspective. In conclusion, while X-ray remains the most commonly used imaging study for diagnosing SLJD,

particularly for ruling out fractures, it can sometimes miss subtle abnormalities, highlighting the need for supplementary imaging techniques.

### US

US is a cost-effective, safe, and efficient method for examining the knee. *Valentino, Quiligotti & Ruggirello (2012)* identified cartilage swelling, patellar tendon thickening, and fragmentation of the lower pole of the patella in 13-year-old football players with SLJD. *Malherbe, 2019* demonstrated that US could detect changes not visible on X-ray, observing apophysis widening with bony irregularity and thickening of the patellar staple at the proximal portion of the patella (*Malherbe, 2019*). *Alito et al. (2023)* found that US and X-rays produced similar results, with US confirming lower pole patella fragmentation, patellar tendon thickening, cartilage swelling, and synovial bursa damage. *De Flaviis et al. (1989)* reported that ultrasound reveals the lower pole of the patella as fragmented and hypoechoic, with significant cartilage swelling, particularly at the patellar tendon insertion (de flaviis). *Carr et al. (2001)* found that US shows thickening of the proximal patellar tendon, with the upper patellar surface appearing smooth while the distal portion is irregular, and a detached bony fragment is visible within the tendon. US seems a highly effective diagnostic tool for SLJD, capable of detecting detailed abnormalities such as cartilage swelling, patellar tendon thickening, and lower pole patella fragmentation, often revealing changes not visible on X-rays.

### MRI

Finally, MRI examination, which is accurate but expensive. Four case studies have reported on MRI findings in patients with SLJD (*Goldmann, 2012*; *Kuehnast, Mahomed & Mistry, 2012*). *Kuehnast, Mahomed & Mistry (2012)* found hyperintense signal in the inferior pole of the patella, in the proximal portion of the patellar staple, and surrounding soft tissues. Similarly, *Goldmann (2012)* described the MRI result as an irregularly shaped lower pole of the patella with an apparent "fracture line" and swelling in an 11-year-old girl. *Davis (2010)* noted that sagittal proton density MRI reveals thickening and increased signal within the proximal patellar tendon just distal to its origin, with an evident intratendinous bone fragment (*Davis, 2010*). *Dupuis et al. (2009)* reported that the corresponding T1-weighted MRI shows low-signal-intensity bone marrow edema, while the second case displays extensive edema at the ossification site and in the surrounding soft tissues (*Dupuis et al., 2009*). MRI provides detailed and precise imaging of SLJD, revealing critical features such as hyperintense signals, irregularly shaped patella, intratendinous bone fragments, and extensive edema that may not be visible through other imaging methods.

## Conservative treatment

The available literature on SLJD primarily consists of case studies with limited descriptions of exact treatment regimens. Despite this, conservative treatment has been the dominant approach, allowing for a natural healing process with varying recovery times (*Franceschi et al., 2007*; *Iwamoto et al., 2009*; *Kuehnast, Mahomed & Mistry, 2012*;

*Valentino, Quiligotti & Ruggirello, 2012*; *Tourdias & Erésué, 2015*; *Alito et al., 2023*). Historically, Sinding-Larson used a plaster dressing with compression on the front of the knee with a fenestra and restricted sports activity. This achieved a positive result after 6 weeks, and return to sports was possible after 6 months in the described patients (*Sinding-Larsen, 1921*). *Bonney (1948)* used plater immoblization (exact information missing) on an 11-year-old football player and after 8 weeks noted fusion of the inferior pattelar osscile and abolition of pain. In contrast, *Kuehnast, Mahomed & Mistry, 2012* reported the case of a 12-year-old boy who was treated with naproxen (250 mg twice a day) and advised to rest from soccer training. The boy reported a withdrawal of symptoms after two months of treatment (*Kuehnast, Mahomed & Mistry, 2012*).

*Alito et al. (2023)* employed NSAID treatment (ibuprofen as needed) and cryotherapy, passive mobilisation of the lower extremities, muscle stretching and isometric exercises (for 8 weeks) for the 10 year-old young soccer player. Subsequently, they implemented muscle-strengthening exercises, including swimming (freestyle and backstroke). Additionally, they advised limiting sports for a minimum of 8–12 weeks. After only 2 months, there was an improvement in function, including improved gait and stair use. After 3 months, an ultrasound showed an improvement in the patellar tendon fibers, and the patient was able to return to activity (*Alito et al., 2023*).

Clinical symptoms and US changes withdrew after 5 months in the process of the natural course of the condition (no information on exact treatment) in the patient of *Valentino, Quiligotti & Ruggirello (2012)*. A young soccer player in a 2015 study by *Tourdias & Erésué (2015)* needed 6 months of conservative treatment (rest, knee splint, muscle stretching) to achieve a favorable functional outcome and return to sports. *López-Alameda et al. (2012)* described a time-injury duration of 1 week to 8 months, (mean, 4 months, large standard deviation = 14 weeks). *Draghi et al. (2008)* described a treatment prognosis of 3–12 months, without showing the evidence or data.

On the other hand, a prospective study by *Materne et al. (2022)* revealed in its study cohort that soccer academy players with a diagnosis of SLJ needed an average of 11 days (11.4 ± 7.2, 1–27 days). These results do not match the data published to date. Perhaps, the players in the *Materne et al. (2022)* study continued to participate in soccer with symptoms (knee pain). A second potential cause may be in the inadequacy of the correct diagnosis of SLJD (*Materne et al., 2022*). Summarizing the case studies described, the time to return to sports activity ranged from 2 to 8 months (*Sinding-Larsen, 1921*; *Kuehnast, Mahomed & Mistry, 2012*; *Valentino, Quiligotti & Ruggirello, 2012*; *López-Alameda et al., 2012*; *Tourdias & Erésué, 2015*; *Alito et al., 2023*).

Considering the amount and strength of scientific evidence, primarily case reports (level five evidence), it may be worth contemplating a procedure akin to conservative treatment for Osgood-Shlatter until new evidence from randomized clinical trials (RCT) becomes available. This suggestion is justified by the similarity of the mechanism of onset, which is traction osteochondrosis, the similar age of patients, and the need for education and modification of physical activity (*Rathleff et al., 2020*).

## Adolescents and children—special considerations

During puberty, young athletes undergo significant biological changes, such as bone growth, that affect their musculoskeletal systemand distinguish them from adult athletes. These changes can increase their susceptibility to sports injuries (*Martínez-Silván et al., 2021*; *Wilczyński et al., 2022a*). It is important to consider personal factors when assessing injury risk (*Hawkins & Metheny, 2001*; *Wilczyński et al., 2022a*). During puberty, there are distinct peak periods where height and weight increase, with a growth spurt occurring between the ages of 10 and 18 years, with a peak value of 8 to 12 cm/year. In boys, growth of bone and lean mass continues until around the age of 18 (*Malina & Bouchard, 1991*; *Hawkins & Metheny, 2001*). Additionally, between the ages of 6 and 14, limb mass increases by a factor of three and lower limb length increases by an average of 1.4 times, resulting in an increase in limb moments of inertia (*Hawkins & Metheny, 2001*).

The rapid changes in body weight and skeletal growth during adolescence place additional stress and fatigue on muscles, tendons, and apophysis. Muscles can adapt relatively quickly to increased demands through hypertrophy and activation of more fibers (*Parker et al., 1990*; *Hawkins & Metheny, 2001*). In contrast, adaptation of the tendon and apophysis of the same muscle can be induced over a longer period of time. This can cause an imbalance, leading to damage to muscle connections during increased loads. Additionally, the increase in muscle force may also be responsible for damaging the tendons and apophysis (*Malina & Bouchard, 1991*; *Hawkins & Metheny, 2001*). Increased tendon loading may contribute to the development of apophysis injuries in young athletes. This may be associated with increased bone and cartilage geometry around the joint, tissue preloading, and decreased flexibility, particularly among boys aged 8 to 13 (*Hawkins & Metheny, 2001*; *Malina et al., 2015*). However, there is still insufficient data on the structural properties of muscle-tendon connections and apophysis, and their relationships with training, maturation, and bone growth in young athletes.

Growth combined with physical activity is linked to a higher incidence of musculoskeletal injuries, including avulsion fractures and apophyseal injuries, with peak growth velocity occurring around ages 10–16 (*Hawkins & Metheny, 2001*). Common lower limb apophyseal injuries, such as Osgood-Schlatter, SLJD, and Sever's disease, occur due to the open areas of growth cartilage in the apophysis (*Hawkins & Metheny, 2001*; *Bruzda, Wilczyński & Zorena, 2023*). This susceptibility is further influenced by the non-linear interaction of limb length, moment of inertia, and body mass during maturation. Young active individuals (at this age) are also at risk of injury due to muscle imbalance, limited neuromuscular control, and improper coordination (*Barber Foss, Myer & Hewett, 2014*; *Wilczyński et al., 2021*, *2022b*). Reduced flexibility found in boys (ages 8 to 13) may also play a role in injury. Factors that can affect injury include increased intensity, duration and volume of physical activity during micro or macro training cycles, poor physical preparation for the season, poor sports training technique, mismatched or lack of protective equipment, lack of warm-up before practice, and inadequate preventive activity. It is important to consider these factors when evaluating the risk of injury (*Patel, Yamasaki & Brown, 2017*; *Prieto-González et al., 2021*).

In summary, tissues that must tolerate dynamic loads need to adapt to higher linear and angular accelerations during non-linear tissue changes. This adaptation involves changes in elasticity, muscle strength, and limb anthropometry. Increased training and match loads, biomechanical issues, and poor physical preparation can predispose individuals to overuse injuries.

## STUDY LIMITATIONS

This review has several limitations that need to be acknowledged. Firstly, there is an inherent selection bias in the inclusion criteria, which may have favored studies that report positive outcomes, potentially skewing the overall findings. This bias arises from a focus on studies that emphasize successful conservative treatments. The reason may also be the reluctance of research authors to publish non-optimistic (or not statistically significant) results. Consequently, the review may present an overly optimistic view of the effectiveness of conservative management for SLJD.

Secondly, the heterogeneity of the included studies, such as variations in study design, patient populations, diagnostic criteria, and treatment protocols, complicates drawing definitive conclusions and developing standardized management approaches for SLJD.

Finally, the scarcity of high-quality, large-scale studies on SLJD limits the robustness of the review's conclusions. A notable limitation of our scoping review is the moderate risk of bias identified in half of the included studies, which often lacked comprehensive details in key areas such as patient history, intervention descriptions, and adverse events. Included studies are case reports or small cohort studies, which necessitates caution when interpreting and applying the findings to broader clinical practice.

In summary, while this review provides valuable insights into the management of SLJD, the potential for selection bias, emphasis on positive outcomes, and the variability and limited scope of the included studies must be considered. Future research should aim to address these limitations by incorporating more rigorous study designs and a balanced representation of findings to enhance the evidence base for managing SLJD.

## CONCLUSIONS

Our review highlight that SLJD predominantly affects physically active male adolescents aged 9–17 years, with common symptoms including insidious pain localized at the inferior pole of the patella, exacerbated by physical activity. Swelling and tenderness around the patellar tendon are common, although knee joint range of motion is usually unaffected. Imaging studies such as X-rays, ultrasound, and MRI play crucial roles in diagnosing SLJD. Recommended conservative treatment ranged from several weeks to several months and included rest and/or cryotherapy, passive mobilization, muscle stretching, isometric exercises, and NSAIDs. Further research involving larger cohorts is essential for refining management protocols and strengthening the evidence base for SLJD treatment.

### Funding

The authors received no funding for this work. The APC was co-financed from the state budget under the program of the Polish Minister of Education and Science under the name "Excellent Science" project no. DNK/SP/548321/2022.

### Grant Disclosures

The following grant information was disclosed by the authors:
Polish Minister of Education and Science: DNK/SP/548321/2022.

### Competing Interests

The authors declare that they have no competing interests.

### Author Contributions

- Bartosz Wilczyński conceived and designed the experiments, performed the experiments, analyzed the data, prepared figures and/or tables, authored or reviewed drafts of the article, and approved the final draft.
- Marcin Taraszkiewicz conceived and designed the experiments, performed the experiments, analyzed the data, authored or reviewed drafts of the article, and approved the final draft.
- Karol de Tillier conceived and designed the experiments, performed the experiments, analyzed the data, authored or reviewed drafts of the article, and approved the final draft.
- Maciej Biały conceived and designed the experiments, performed the experiments, analyzed the data, authored or reviewed drafts of the article, and approved the final draft.
- Katarzyna Zorena conceived and designed the experiments, performed the experiments, analyzed the data, authored or reviewed drafts of the article, and approved the final draft.

### Data Availability

The data (complete search strategy, JBI quality of included studies and final protocol) that support the findings of this study are available in Open Science Framework Wilczyński, Bartosz. 2024. "Sinding-Larsen-Johansson Disease. Clinical Features, Imaging Findings, Conservative Treatments and Research Perspectives. A Scoping Review." OSF. August 15. https://osf.io/z7cnw/.

### Supplemental Information

Supplemental information for this article can be found online at http://dx.doi.org/10.7717/peerj.17996#supplemental-information.

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
