# Peer review of "Sinding-Larsen-Johansson disease. Clinical features, imaging findings, conservative treatments and research perspectives: a scoping review"

_PeerJ, doi:10.7717/peerj.17996_

## Round 0.1 · original submission · Major Revisions

· Academic Editor

Major Revisions

Dear Dr. Wilczyński,

Your manuscript titled "Sinding-Larsen-Johansson disease. Clinical features, Imaging findings, Conservative treatments and Research perspectives: a Scoping Review" was considered by two expert reviewers and based on their opinions and my review, the decision is “Major Revisions”.

(1) Note that the reviewers were positive in their comments but at the same time felt that the manuscript requires more work before it can be accepted.
(2) Both reviewers’ commented on the language and syntax of the manuscript. Please carefully read the manuscript and simply convoluted and unclear sentences.
(3) Please also address comments about figure 1 and table 1 legends.
(4) Methodology needs to be more detailed and transparent. Please reads and address reviewers’ comments carefully (& add the date the web search was performed).
(5) Both reviewers indicated that more references (especially more recent ones) are missing.

Please carefully read the reviewers’ comments and address them fully in your revised manuscript. In addition, please address the following points:
- L83: “… the casue of the traction mechanism is similar to Osgood-Shlatter disease …”. “cause” and not “cause”. In addition, please add a sentence or two explaining the traction mechanism.
- L90: “The second hypothesis ...”. What is the first hypothesis?
- L96: “… 12.5% among all athletes …”. Thin is not correct. It is only among soccer players.
- L109-10: “According to ... concurrent OSD”. This sentence needs to be rewritten as it lacks the correct syntax. Furthermore, this sentence is repeated twice (L110-11).
- L114: delete “study,”
- L148-9: using keywords: "Sinding Larsen-Johansson", "traction osteochondroses", "knee apophysis", "knee apophysitis", with Boolean operator "AND". This is not clear to me. The term “or” was supposed to be used. Using “and” will only include articles that have all 4 terms. Please also add the date of the search.
- L310-12: “… open areas of growth cartilage in the apophysis …”. This statement repeats twice in consecutive sentences.
-Figure 1: “record removed for other reasons”. Please specify.

Please ensure that all review, editorial, and staff comments are addressed in a response letter and any edits or clarifications mentioned in the letter are also inserted into the revised manuscript where appropriate.

Please note that submitting a revision of your manuscript does not guarantee eventual acceptance, and that your revision may be subject to re-review by the reviewer(s) before a decision is rendered.

Reviewer 1 ·

Basic reporting

a. Language and Clarity: Several sentences could be more transparent and more accessible to follow, detracting from the overall clarity of the manuscript. For Example: "SLJD, while predominantly a condition affecting adolescents, with the characteristic feature being the pain localized at the inferior pole of the patella, often resulting in difficulty in diagnosis due to overlapping symptoms with other knee pathologies."

b. Introduction and Background: The introduction needs to adequately frame the significance of SLJD in the context of recent literature. For Example:** The review lacks recent references, particularly from the last five years, to underscore current research trends and advancements. Include recent studies and reviews, highlighting advances in diagnostic and treatment techniques, placing the review in a contemporary context, and giving other specialty reviews. A cursory search on Pubmed yields numerous examples.

c. Structure and Figures: Figures and tables need more detailed descriptions, reducing their effectiveness. Figure 1 is poorly labeled and lacks a comprehensive legend.

Experimental design

a. Scope and Originality: The scope needs to be broader, leading to superficial treatment of critical issues. The review attempts to cover clinical features, imaging, treatments, and research perspectives but needs to provide an in-depth analysis of each topic. Narrow the scope for a more detailed and nuanced discussion of critical aspects, such as focusing specifically on imaging techniques or treatment efficacy.
b. Methodology: The Methodology needs more transparency in search strategies and selection processes. The exact search terms and databases used are not disclosed, which limits Reproducibility. Provide a detailed account of the search strategy, including specific databases, search terms, and inclusion/exclusion criteria.
c. Reproducibility: The need for detailed methodological information hampers Reproducibility.

Validity of the findings

a. Data Robustness and Analysis: The review needs to critically appraise the quality of included studies, leading to potential bias. Studies of varying quality are presented without sufficient implementation of a standardized tool for quality assessment of included studies, and the potential impact of study quality on the findings is discussed.

b. Conclusions: The conclusions are speculative and not fully supported by robust evidence. Recommendations for specific treatments are made despite limited, low-quality evidence. Base conclusions on high-quality evidence and acknowledge the limitations of the available data. Only make definitive recommendations when evidence is strong or consistent.

Additional comments

The manuscript addresses an important and clinically relevant topic, aiming to synthesize current knowledge on SLJD.
The review often highlights studies that support a particular viewpoint while neglecting those with contradictory findings. Positive outcomes of conservative treatments are emphasized without adequately discussing studies showing limited or no benefit.
The inclusion criteria favor certain studies, potentially skewing the findings. The review disproportionately includes studies with positive findings, which may represent only part of the available research spectrum.

Reviewer 2 ·

Basic reporting

The article addresses a significant and relatively understudied topic in pediatric sports medicine, providing a comprehensive overview of Sinding-Larsen-Johansson Disease (SLJD). However, the review would benefit from further clarity and depth in some sections, particularly in the methodology and discussion.However, there are areas where the manuscript can be improved for better clarity and impact.

Article covers an important topic but needs more clarity and depth in methodology and discussion

Experimental design

The methods section mentions using the PRISMA-ScR framework, which is appropriate. However, the search strategy and eligibility criteria could be described in more detail. For instance, what were the specific inclusion and exclusion criteria for the studies? How was the quality of the included studies assessed, if at all?

The process of data extraction and synthesis could also be elaborated upon. How were discrepancies between reviewers resolved? Were any tools or software used for managing and analyzing the data?


More information on the inclusion and exclusion criteria is needed. For example, what were the reasons for excluding certain studies? How was the relevance and quality of studies assessed?

Validity of the findings

Article mention that best of our current knowledge, this is the first review strictly focused of SLJD research so what's the LEARNING POINT OF VIEW in the article helping clinicians to early diagnosis of disease


Please give protocol to help to reach diagnosis

Additional comments

Article says Limitations of this review include the lack of quality assessment of the included studies or in-depth analysis of the effectiveness of the interventions. Next limitation is the low number of included studies. In our study, we focused on young participants with diagnosed SLJD, no comorbidities, and no consequences of SLJD (such as patella fractures).

THEY SHOULD USE REFERENCE BOOKS AND SYMPOSIUM

Shorten the abstract, ensure references are current, and proofread for error and include recent studies, and simplify complex sentences.



The conclusion is comprehensive but somewhat repetitive. Streamline this section to focus on the main takeaways and the most important recommendations for practice and research.

---

## Round 0.2 · Minor Revisions

· Academic Editor

Minor Revisions

Dear Dr. Wilczyński,

Your manuscript titled "Sinding-Larsen-Johansson disease. Clinical features, Imaging findings, Conservative treatments and Research perspectives: a Scoping Review" was reconsidered by an expert reviewer and based on their opinion and my review, the decision is “Minor Revisions”.

Please note that the legend for figure 1 is repeating the main text ad verbum. This is redundant. Please rewrite the legend to describe the figure (a flowchart describing the selection process of suitable papers included in this review…).

Please ensure that all review, editorial, and staff comments are addressed in a response letter and any edits or clarifications mentioned in the letter are also inserted into the revised manuscript where appropriate.

Please note that submitting a revision of your manuscript does not guarantee eventual acceptance, and that your revision may be subject to re-review by the reviewer(s) before a decision is rendered.

Reviewer 2 ·

Basic reporting

CHANGES MADE ARE OKAY

Experimental design

OKAY

Validity of the findings

LOOK OK PUBLISH

---

## Round 0.3 · accepted · Accept

· Academic Editor

Accept

Dear Dr. Wilczyński,

Thank you for submitting your revised manuscript titled "Sinding-Larsen-Johansson disease. Clinical features, Imaging findings, Conservative treatments and Research perspectives: a A Scoping Review". After reading the revised manuscript I’m happy to let you know that the decision is “accept”.